# Are perioperative interventions effective in preventing chronic pain after primary total knee replacement? A systematic review

Andrew David Beswick,[1] Jane Dennis,[1] Rachael Gooberman-Hill,[1,2] Ashley William Blom,[1,2] Vikki Wylde[1,2]

¹Musculoskeletal Research Unit, Translational Health Sciences, Bristol Medical School, University of Bristol, Bristol, UK
²National Institute for Health Research Bristol Biomedical Research Centre, University Hospitals Bristol NHS Foundation Trust and University of Bristol, Bristol, UK

**Correspondence to**
Andrew David Beswick;
andy.beswick@bristol.ac.uk

## ABSTRACT

**Objectives** For many people with advanced osteoarthritis, total knee replacement (TKR) is an effective treatment for relieving pain and improving function. Features of perioperative care may be associated with the adverse event of chronic pain 6 months or longer after surgery; effects may be direct, for example, through nerve damage or surgical complications, or indirect through adverse events. This systematic review aims to evaluate whether non-surgical perioperative interventions prevent long-term pain after TKR.

**Methods** We conducted a systematic review of perioperative interventions for adults with osteoarthritis receiving primary TKR evaluated in a randomised controlled trial (RCT). We searched *The Cochrane Library*, MEDLINE, Embase, PsycINFO and CINAHL until February 2018. After screening, two reviewers evaluated articles. Studies at low risk of bias according to the Cochrane tool were included.

**Interventions** Perioperative non-surgical interventions; control receiving no intervention or alternative treatment.

**Primary and secondary outcome measures** Pain or score with pain component assessed at 6 months or longer postoperative.

**Results** 44 RCTs at low risk of bias assessed long-term pain. Intervention heterogeneity precluded meta-analysis and definitive statements on effectiveness. Good-quality research provided generally weak evidence for small reductions in long-term pain with local infiltration analgesia (three studies), ketamine infusion (one study), pregabalin (one study) and supported early discharge (one study) compared with no intervention. For electric muscle stimulation (two studies), anabolic steroids (one study) and walking training (one study) there was a suggestion of more clinically important benefit. No concerns relating to long-term adverse events were reported. For a range of treatments there was no evidence linking them with unfavourable pain outcomes.

**Conclusions** To prevent chronic pain after TKR, several perioperative interventions show benefits and merit further research. Good-quality studies assessing long-term pain after perioperative interventions are feasible and necessary to ensure that patients with osteoarthritis achieve good long-term outcomes after TKR.

### Strengths and limitations of this study

► For the first time, this systematic review brings together contemporary evidence on aspects of perioperative care for people with total knee replacement and their effects on long-term pain.
► Only studies assessed to be at low risk of bias were included in the narrative synthesis.
► Intervention and outcome heterogeneity precluded meta-analysis.

## BACKGROUND

In the USA about 13% of men and 19% of women will be diagnosed with knee osteoarthritis and over half will receive a total knee replacement (TKR).[1] For people with advanced osteoarthritis unresponsive to pharmacological or conservative treatments, TKR aims to relieve pain and improve function. In the UK, nearly 100 000 primary TKRs were performed in 2017[2,3] and in the USA in 2010, an estimated 4.7 million people were living with a TKR.[4] Despite good outcomes for many, some people report long-term pain and are disappointed with their surgery.[5,6] After TKR, pain levels plateau from about 6 months[7,8] after which persistent pain is considered 'chronic'[9] and is reported by 10%–34% of patients.[10]

The mechanisms that influence the development of chronic pain after TKR may be biological, mechanical and psychosocial. Biological explanations include the sensitising impact of long-term pain from osteoarthritis,[11,12] inflammation, infection and localised nerve injury.[13] Mechanical explanations include altered gait, prosthesis loosening and effects on ligaments.[14,15] Psychological factors including depression and catastrophising may also influence outcomes.[16–19] Much research has focused on preoperative predictors of outcomes and

these include pain intensity, presence of widespread pain, anxiety, depression and catastrophising.[10] [20] However, attempts to target or modify preoperative care have, as yet, shown no benefit regarding chronic pain or other long-term patient outcomes.[10] [21–23]

Perioperative risk factors suggest that appropriate interventions may reduce long-term pain. For example, acute postoperative pain, which may be a direct consequence of the operation, anaesthetic protocol and subsequent analgesia, or related to particular aspects of care, is an acknowledged risk factor for chronic postsurgical pain.[24]

In the perioperative period from hospital admission to the early stages of recovery, care focuses on acute pain management, prevention of adverse events, facilitation of early mobilisation and timely discharge. However, for people with osteoarthritis the key aim of TKR is the achievement of a long-term painless and well-functioning knee with no adverse events. All aspects of perioperative care should work together to achieve this.

Any treatment in the perioperative period including pain management, blood conservation, deep vein thrombosis (DVT) and infection prevention, and inpatient rehabilitation could potentially affect patient recovery and chronic pain, either directly or indirectly. Direct mechanisms may be through prevention of nerve damage,[25] post-thrombotic syndrome,[26] reperfusion injury[27] and articular bleeding.[28] For other treatments, pathways leading to long-term pain may be indirect, possibly being mediated through increased risks of adverse events.[29] Irrespective of mechanism, chronic pain is a highly prevalent adverse event after TKR and should be considered along with infection, DVT and other complications in the safety profile of interventions.

Our systematic review of randomised controlled trials (RCT) aims to evaluate the effectiveness of treatments in the perioperative period in preventing long-term pain after TKR. By focusing on studies with low risk of bias we aim to identify interventions with robust evidence of long-term effectiveness and identify gaps in the research base.

## METHODS

The systematic review protocol was registered (PROSPERO CRD42017041382) and PRISMA reporting guidelines used.[30] A checklist is included as online supplementary material.

### Patient and public involvement

As part of the STAR programme of research (NIHR RP-PG-0613-20001), this review benefited from extensive patient and public involvement. Advice was sought from patients and stakeholders at a group discussion in March 2016 with decisions made on inclusion criteria and outcomes. Our patient advisory group comprises five patients with experience of long-term pain after TKR, supported by a dedicated coordinator. This group will advise on dissemination of the study results to a general audience including plain language summaries.

### Eligibility criteria

Studies were eligible if they satisfied Population-Intervention-Comparator-Outcome Study criteria defined in the protocol. Participants were adults receiving unilateral primary TKR with osteoarthritis in at least 75% of patients. Pharmacological or non-pharmacological interventions commenced in the perioperative setting with 'peri-operative' reflecting the time from hospital admission to immediately postdischarge. Interventions relating to implant designs and surgical procedures were excluded. The comparator was usual care, placebo or an alternative intervention. Outcomes were, in preference, patient-reported joint-specific pain intensity measured by tools such as the Western Ontario and McMaster Universities Osteoarthritis Index (WOMAC) or Oxford Knee Score (OKS). If joint-specific measures were unavailable, pain dimensions from quality of life measures were used or pain rated on a visual analogue scale (VAS) or numerical rating scale (NRS). We also considered composite patient-reported outcome measures and surgeon scores which included a pain intensity component, such as the American Knee Society Score (KSS) and Hospital for Special Surgery (HSS) score. Measures specifically of neuropathic pain were also used. The occurrence of adverse events was summarised. The studies included were RCTs with follow-up at ≥6 months after surgery and a pain outcome or score including pain. Authors of studies were contacted regarding incomplete pain outcome data.

### Database searches

We established an Endnote database of all RCTs in TKR. On 14 February 2018, a search from database inception was conducted in: *The Cochrane Library*; MEDLINE, Embase and PsycINFO on Ovid; and CINAHL on EBSCOhost. The MEDLINE search strategy is included as online supplementary material. Citations of key articles were tracked in Web of Science. No language restrictions were applied, and translations made. Studies reported as abstracts or unobtainable using interlibrary loans and author contact were excluded.

### Screening and data extraction

We imported records into Endnote X7 (Thomson Reuters). An initial screen by one reviewer excluded clearly irrelevant articles. Subsequently, abstracts and full articles were screened independently by two reviewers and reasons for exclusion recorded.

Data were extracted onto piloted forms and an Excel spreadsheet by one reviewer, specifically: country; dates; participants (indication, age, sex); inclusion and exclusion criteria; intervention and control content; setting, timing, duration and intensity of intervention; follow-up intervals; losses to follow-up; pain outcome data; and serious adverse events. Data were checked against source material by a second reviewer.

Authors were contacted for missing data, and data provided for previous reviews were used.[10] [31]

## Quality assessment

Potential sources of bias were assessed by two experienced reviewers using the Cochrane risk of bias tool,[32] specifically: the randomisation process; deviations from intended interventions; missing outcome data (>20%), measurement of the outcome; and selection of the reported result. Studies with serious concerns relating to risk of bias were considered high risk and those with limited reporting unclear risk. Studies with high or unclear risk of bias were excluded from the narrative synthesis but are included in online supplementary summary tables with reasons for exclusion.

## Data analysis

Insufficient studies with similar interventions and outcomes were identified for meta-analysis, and a narrative synthesis is presented. Results reported with p values ≤0.001 were considered 'strong' evidence of effectiveness,[33] p values 0.001–0.05 'some' evidence and p values 0.05–0.1 'weak' evidence. When authors reported results 'statistically significant' with no p value, this was noted. Where possible, effect sizes were compared with published minimal clinically important differences (MCID). Concerns relating to adverse events were summarised.

## RESULTS

Figure 1 shows review progress and reasons for exclusion. Of 1515 RCTs of interventions in the perioperative setting, 1385 had no long-term follow-up. Perioperative interventions with follow-up of ≥6 months were evaluated in 130 RCTs of which 76 reported a pain outcome or score with a pain component. Detailed intervention and study characteristics and risk of bias assessments are provided as online supplementary material. Studies excluded had concerns for risk of bias pertaining to at least one of: large baseline differences in group characteristics or numbers in groups (n=4); incomplete outcome data (n=15); limited or selective reporting (n=12); or unblinded surgeon follow-up (n=1).

Details of 44 studies assessed to be at low risk of bias are summarised in table 1. In 34 studies, patients received TKR exclusively for osteoarthritis and in three studies, 75% or more patients had osteoarthritis. In seven studies there was no information on reason for surgery but there was no suggestion that patients had an indication other than osteoarthritis. Interventions focused on pain management (n=20), tourniquets (n=5), compression bandages (n=1), blood conservation (n=7), denosumab (n=1), continuous passive motion (CPM, n=2), electrical stimulation (n=2), rehabilitation (n=4), wound management (n=1) and anabolic steroids (n=1). Primary pain outcome measures reported were VAS or NRS pain (n=12), WOMAC pain (n=7), Knee injury and Osteoarthritis Outcome Score (KOOS) pain (n=3), Leeds Assessment of Neuropathic Symptoms and Signs Pain Scale (S-LANSS) (n=1), Short Form 36 (SF-36) bodily pain (n=1), or composite scores including a pain measure,

OKS or WOMAC (n=10), KSS or HSS (n=10). Latest outcomes were recorded at 6 months (n=12), 12 months (n=26) and 24 months (n=6). Reporting of adverse events covered the entire follow-up period in 27 studies, short term after surgery in 15 studies, but were not reported in two studies.

## Pain management

We identified 20 RCTs with 2393 participants evaluating components of multimodal pain management. Four studies each were from China and the USA, two each from Canada and the UK and one each from Australia, Finland, France, Iran, Singapore, South Korea, Sweden and The Netherlands. All were conducted at a single centre and, in those with dates, participants were recruited between 2004 and 2015. Sample sizes ranged from 44 to 280 participants, with a median of 96. Four studies had three trial arms and 16 had two. The range of mean or median ages of participants in randomised groups was 61–73 years and, in 17/19 studies with data, a majority of participants were women.

### Femoral nerve block

Femoral nerve blocks (FNB) were studied in 10 RCTs.

Three RCTs compared FNB with no FNB. In one study with 55 patients, WOMAC pain scores at 1 year were similar in patients receiving single-shot FNB and untreated controls.[34] All patients received local infiltration analgesia (LIA) and patient-controlled analgesia (PCA). In another study with all participants receiving LIA, 150 were randomised to receive single-shot FNB with or without sciatic nerve block (SNB), or general anaesthesia.[35] There were no differences in HSS scores between groups at 6 months. Continuous FNB was compared with oral hydrocodone opioid in 62 patients receiving PCA.[36] There was some evidence for 'pain using stairs' favouring hydrocodone (p=0.01) but no difference in overall NRS-rated pain at 1 year and concern over venous thromboembolism in 4/31 participants treated with hydrocodone.

In two RCTs, continuous FNB was compared with PCA. In one study with 60 participants, the KSS at 6 months was similar between groups.[37] In another study with 280 participants, there was some evidence for higher incidence of NRS-rated pain at 6 months in the PCA group than the FNB group (p=0.021) but not at 12 months (p=0.273).[38]

Two RCTs compared FNB with LIA. In one study, all 157 participants also received PCA.[39] At 1 year, KSS values were similar in single-shot FNB and LIA groups. In the other study, 94 participants were randomised to receive single-shot FNB with continuous epidural infusion or LIA through an intra-articular catheter.[40] VAS-rated pain was similar between groups at 1 year.

In two RCTs, FNB procedures were compared. In one study with 99 patients randomised to two FNB concentrations, there was no difference in WOMAC score between groups at 12 months.[41] In another study with 61 participants allocated to two different durations of FNB, there was no difference in WOMAC pain scores at 1 year.[42] In

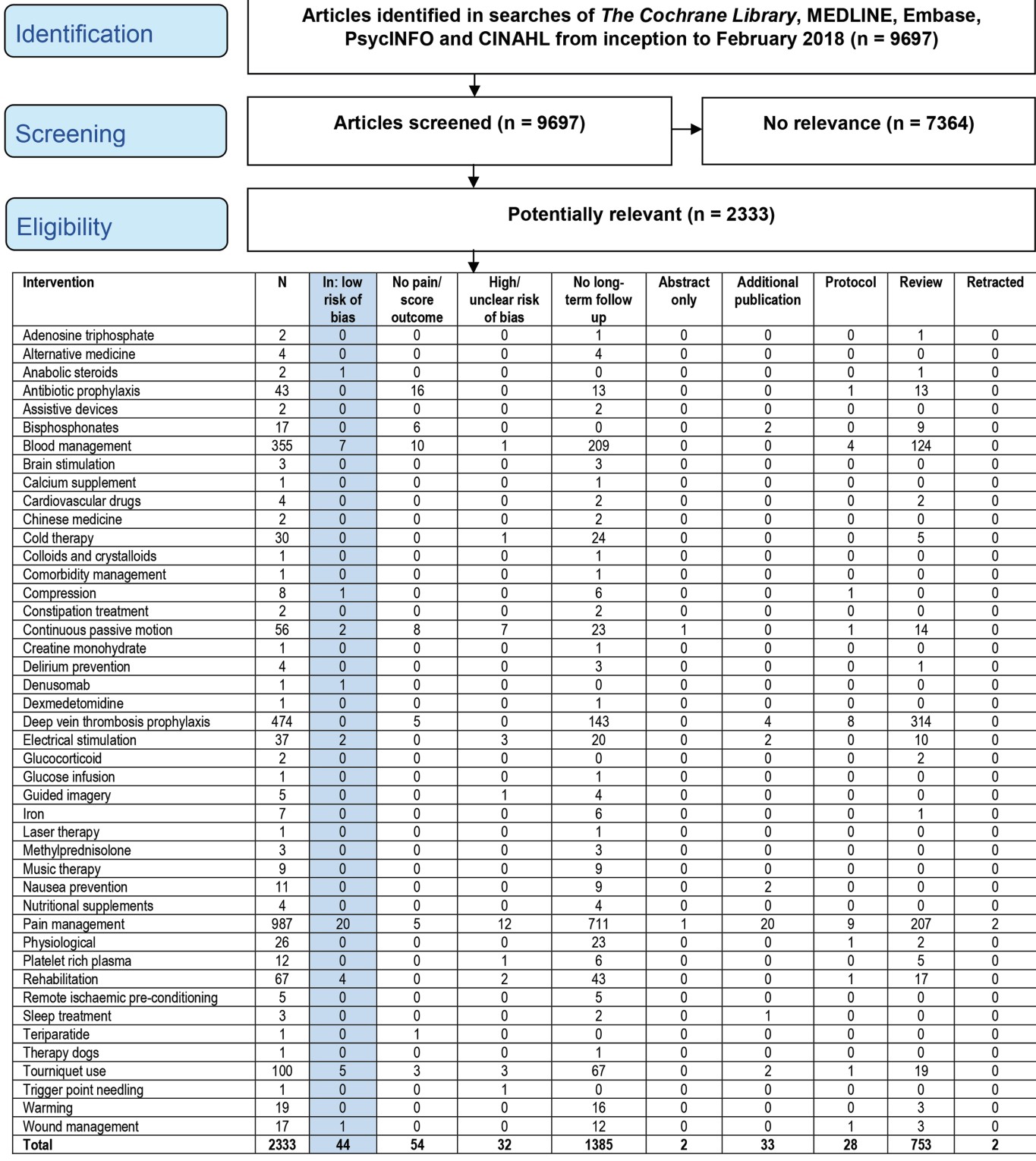

| Intervention | N | In: low risk of bias | No pain/ score outcome | High/ unclear risk of bias | No long-term follow up | Abstract only | Additional publication | Protocol | Review | Retracted |
|---|---|---|---|---|---|---|---|---|---|---|
| Adenosine triphosphate | 2 | 0 | 0 | 0 | 1 | 0 | 0 | 0 | 1 | 0 |
| Alternative medicine | 4 | 0 | 0 | 0 | 4 | 0 | 0 | 0 | 0 | 0 |
| Anabolic steroids | 2 | 1 | 0 | 0 | 0 | 0 | 0 | 0 | 1 | 0 |
| Antibiotic prophylaxis | 43 | 0 | 16 | 0 | 13 | 0 | 0 | 1 | 13 | 0 |
| Assistive devices | 2 | 0 | 0 | 0 | 2 | 0 | 0 | 0 | 0 | 0 |
| Bisphosphonates | 17 | 0 | 6 | 0 | 0 | 0 | 2 | 0 | 9 | 0 |
| Blood management | 355 | 7 | 10 | 1 | 209 | 0 | 0 | 4 | 124 | 0 |
| Brain stimulation | 3 | 0 | 0 | 0 | 3 | 0 | 0 | 0 | 0 | 0 |
| Calcium supplement | 1 | 0 | 0 | 0 | 1 | 0 | 0 | 0 | 0 | 0 |
| Cardiovascular drugs | 4 | 0 | 0 | 0 | 2 | 0 | 0 | 0 | 2 | 0 |
| Chinese medicine | 2 | 0 | 0 | 0 | 2 | 0 | 0 | 0 | 0 | 0 |
| Cold therapy | 30 | 0 | 0 | 1 | 24 | 0 | 0 | 0 | 5 | 0 |
| Colloids and crystalloids | 1 | 0 | 0 | 0 | 1 | 0 | 0 | 0 | 0 | 0 |
| Comorbidity management | 1 | 0 | 0 | 0 | 1 | 0 | 0 | 0 | 0 | 0 |
| Compression | 8 | 1 | 0 | 0 | 6 | 0 | 0 | 1 | 0 | 0 |
| Constipation treatment | 2 | 0 | 0 | 0 | 2 | 0 | 0 | 0 | 0 | 0 |
| Continuous passive motion | 56 | 2 | 8 | 7 | 23 | 1 | 0 | 1 | 14 | 0 |
| Creatine monohydrate | 1 | 0 | 0 | 0 | 1 | 0 | 0 | 0 | 0 | 0 |
| Delirium prevention | 4 | 0 | 0 | 0 | 3 | 0 | 0 | 0 | 1 | 0 |
| Denusomab | 1 | 1 | 0 | 0 | 0 | 0 | 0 | 0 | 0 | 0 |
| Dexmedetomidine | 1 | 0 | 0 | 0 | 1 | 0 | 0 | 0 | 0 | 0 |
| Deep vein thrombosis prophylaxis | 474 | 0 | 5 | 0 | 143 | 0 | 4 | 8 | 314 | 0 |
| Electrical stimulation | 37 | 2 | 0 | 3 | 20 | 0 | 2 | 0 | 10 | 0 |
| Glucocorticoid | 2 | 0 | 0 | 0 | 0 | 0 | 0 | 0 | 2 | 0 |
| Glucose infusion | 1 | 0 | 0 | 0 | 1 | 0 | 0 | 0 | 0 | 0 |
| Guided imagery | 5 | 0 | 0 | 1 | 4 | 0 | 0 | 0 | 0 | 0 |
| Iron | 7 | 0 | 0 | 0 | 6 | 0 | 0 | 0 | 1 | 0 |
| Laser therapy | 1 | 0 | 0 | 0 | 1 | 0 | 0 | 0 | 0 | 0 |
| Methylprednisolone | 3 | 0 | 0 | 0 | 3 | 0 | 0 | 0 | 0 | 0 |
| Music therapy | 9 | 0 | 0 | 0 | 9 | 0 | 0 | 0 | 0 | 0 |
| Nausea prevention | 11 | 0 | 0 | 0 | 9 | 0 | 2 | 0 | 0 | 0 |
| Nutritional supplements | 4 | 0 | 0 | 0 | 4 | 0 | 0 | 0 | 0 | 0 |
| Pain management | 987 | 20 | 5 | 12 | 711 | 1 | 20 | 9 | 207 | 2 |
| Physiological | 26 | 0 | 0 | 0 | 23 | 0 | 0 | 1 | 2 | 0 |
| Platelet rich plasma | 12 | 0 | 0 | 1 | 6 | 0 | 0 | 0 | 5 | 0 |
| Rehabilitation | 67 | 4 | 0 | 2 | 43 | 0 | 0 | 1 | 17 | 0 |
| Remote ischaemic pre-conditioning | 5 | 0 | 0 | 0 | 5 | 0 | 0 | 0 | 0 | 0 |
| Sleep treatment | 3 | 0 | 0 | 0 | 2 | 0 | 1 | 0 | 0 | 0 |
| Teriparatide | 1 | 0 | 1 | 0 | 0 | 0 | 0 | 0 | 0 | 0 |
| Therapy dogs | 1 | 0 | 0 | 0 | 1 | 0 | 0 | 0 | 0 | 0 |
| Tourniquet use | 100 | 5 | 3 | 3 | 67 | 0 | 2 | 1 | 19 | 0 |
| Trigger point needling | 1 | 0 | 0 | 1 | 0 | 0 | 0 | 0 | 0 | 0 |
| Warming | 19 | 0 | 0 | 0 | 16 | 0 | 0 | 0 | 3 | 0 |
| Wound management | 17 | 1 | 0 | 0 | 12 | 0 | 0 | 1 | 3 | 0 |
| Total | 2333 | 44 | 54 | 32 | 1385 | 2 | 33 | 28 | 753 | 2 |

**Figure 1** Systematic review flow diagram.

these studies, all participants received either SNB[41] or PCA.[42]

Single-shot FNB was compared with single adductor canal block in one RCT with 98 participants, all receiving LIA.[43] At 6 months there was no difference in VAS-rated pain.

### Sciatic nerve block

In one study, 89 patients were randomised to single-shot SNB, continuous SNB, or PCA.[44] All patients received FNB. At 12 months, there were no differences in pain for single-shot SNB and continuous SNB on the WOMAC pain scale or VAS-rated pain at rest or during mobilisation.

**Table 1** Perioperative interventions with follow-up for pain or score at 6 months or later and assessed to be at low risk of bias

| Study, country, patient recruitment dates Setting | Treatment common to randomised groups | Intervention | Patients, n | Follow-up Group difference |
|---|---|---|---|---|
| **Pain management: nerve blocks** | | | | |
| Albrecht et al[41] Canada, 2009–2011 1 hospital | SNB | 1. FNB continuous high 2. FNB continuous low 3. FNB single | 99 | 1 year WOMAC score: no difference (p=0.68) |
| Choy et al[42] Korea, 2006–2007 1 surgeon | PCA | 1. FNB continuous long 2. FNB continuous short | 61 | 1 year WOMAC pain: no difference (p=0.2) |
| Fan et al[39] China, 2012–2014 2 surgeons | PCA | 1. FNB single 2. LIA | 157 | 1 year KSS: no difference (p=0.51) |
| Gao et al[35] China, 2014–2015 1 centre | LIA | 1. General anaesthesia 2. FNB single 3. FNB/SNB single | 150 | 6 months HSS score: no significant difference (p>0.05) |
| Macrinici et al[43] USA, before 2017 1 centre | LIA | 1. ACB single 2. FNB single | 98 | 6 months VAS pain: no difference |
| Nader et al[36] USA, 2007–2008 1 surgeon | PCA | 1. FNB continuous 2. Oral opioid | 62 | 1 year NRS pain stair: some evidence favouring opioid (p=0.01) but not consistent. Overall NRS pain: no difference (p=1.0) VTE: concern opioid |
| Peng et al[38] China, Before 2014 1 centre | | 1. FNB continuous 2. PCA | 280 | 6 months and 1 year NRS pain: some evidence favouring FNB at 6 months (p=0.021); no difference at 1 year (p=0.273) |
| Reinhardt et al[40] USA, 2010–2012 2 surgeons | | 1. FNB single/ epidural 2. LIA 48 hours | 94 | 1 year VAS pain: no difference |
| Wegener et al[44] The Netherlands, 2008–2010 1 centre | FNB | 1. SNB single 2. SNB continuous 3. PCA | 89 | 1 year WOMAC pain: no difference (p=0.81) |
| Widmer et al[34] Australia, before 2012 2 surgeons | LIA, PCA | 1. FNB single 2. Control no FNB | 55 | 1 year WOMAC pain: no difference (p=0.74) |
| Wu and Wong[37] China, 2009–2011 1 centre | | 1. FNB continuous 2. PCA | 60 | 6 months KSS: no difference (p=0.513) |
| **Pain management: LIA** | | | | |
| McDonald et al[52] UK, 2010–2011 1 hospital | | 1. LIA 2. PCA | 222 | 1 year OKS: no difference (p=0.915) |
| Motififard et al[49] Iran, 2014–2015 1 hospital | | 1. LIA pre-emptive injection 2. Control saline with epinephrine | 120 | 6 months KSS: weak evidence favouring LIA (p=0.07). Difference between groups (14.2/200) less than MCID (12.3/200) |
| Niemeläinen et al[47] Finland, 2011–2012 1 hospital | PCA | 1. LIA 2. Control saline | 56 | 1 year OKS: weak evidence from means and CIs favouring LIA. Difference (2.7/48) less than MCID (4.0/48) |
| Sean et al[53] Singapore, 2004–2005 1 hospital | PCA | 1. LIA with corticosteroid 2. LIA no corticosteroid | 100 | 6 months and 2 years OKS: no difference |

**Table 1** Continued

| Study, country, patient recruitment dates Setting | Treatment common to randomised groups | Intervention | Patients, n | Follow-up Group difference |
|---|---|---|---|---|
| Williams et al[51] Canada, Before 2013 2 surgeons | LIA, PCA | 1. LIA 48 hours 2. Control saline | 51 | 6 months and 1 year VAS pain: no difference (6 months p=0.836, 1 year p=0.767) |
| Wylde et al[45] UK, 2009–2012 1 centre | FNB, PCA | 1. LIA 2. Control no LIA | 280 | 6 months and 1 year WOMAC pain: weak evidence favouring LIA at 6 months p=0.063; 1 year p=0.107. Mean difference at 1 year (3.8/100) lower than MCID (8–9/100) |
| **Pain management: celecoxib** | | | | |
| Meunier et al[54] Sweden, 2004–2005 1 centre | PCA | 1. Celecoxib 2. Control placebo | 44 | 1 year KOOS/VAS pain: no difference |
| **Pain management: ketamine/nefopam** | | | | |
| Aveline et al[55] France, 2005 1 centre | PCA | 1. Ketamine infusion 2. Nefopam infusion 3. Control saline | 75 | 6 months and 1 year DN4/VAS pain: some evidence favouring ketamine (for DN4 p=0.02). Few patients had neuropathic pain at 12 months. |
| **Pain management: pregabalin** | | | | |
| Buvanendran et al[56] USA, 2006–2007 Single centre | LIA, PCA | 1. Pregabalin 2. Control placebo | 240 | 6 months NRS pain: some evidence favouring pregabalin at 6 months (p=0.0176) S-LANSS pain: no neuropathic pain reported in pregabalin group compared with 5.2% of patients in control group (p=0.014) Sedation and confusion day 0 and day 1: concern pregabalin |
| **Tourniquet** | | | | |
| Ejaz et al[58] Denmark, 2011–2012 1 centre | Tranexamic acid | 1. Tourniquet 2. Tourniquet not inflated | 64 | 6 months and 1 year KOOS pain: no significant difference |
| Huang et al[60] China, 2015 1 centre | Tranexamic acid | 1. Tourniquet 2. No tourniquet | 100 | 6 months VAS pain: no difference (p=0.728) Wound: concern tourniquet |
| Liu et al[59] Australia, Before 2014 1 surgeon | | 1. Tourniquet 2. Tourniquet not inflated | 20 | 6 months and 1 year OKS: no significant difference Transfusion: concern tourniquet |
| Mittal et al[61] Australia, 2008–2010 1 centre | | 1. Tourniquet short duration 2. Tourniquet long duration | 65 | 1 year OKS: weak evidence from means and CIs on graph favouring long duration at 1 year. Mean difference (5) greater than MCID (4) Transfusions/adverse events: concern short |

Continued

**Table 1** Continued

| Study, country, patient recruitment dates Setting | Treatment common to randomised groups | Intervention | Patients, n | Follow-up Group difference |
|---|---|---|---|---|
| Zhang et al[62] China, 2008–2011 1 surgeon | | 1. Tourniquet for entire operation 2. Tourniquet removed before wound closure 3. Tourniquet from first bone osteotomy until closure | 150 | 6 months HSS score: no difference (p=0.839) Transfusions: concern late tourniquet start in groups 1 and 2 |
| **Compression bandage** | | | | |
| Brock et al[70] UK, 2013–2014 1 hospital | Hydrocolloid dressing | 1. Compression bandage 2. Standard crepe bandage | 49 | 6 months OKS: no difference (p=0.58) |
| **Blood conservation** | | | | |
| Hourlier et al[67] France, 2009–2010 1 hospital | Drain, tourniquet, electrocautery | 1. Continuous infusion tranexamic acid 2. Control saline | 106 | 6 months KSS: no difference (p=0.90) |
| Huang et al[60] China, 2015 1 centre | Tourniquet | 1. Intravenous and topical tranexamic acid 2. No tranexamic acid | 100 | 6 months VAS pain: no difference (p=0.728) HSS score: strong evidence favouring tranexamic acid (p<0.001). Mean difference (1.4/100) lower than MCID (8.3/100) Blood loss: control concern |
| Kim et al[63] Korea, 2009–2011 1 hospital | Tourniquet, drain, compressive dressing | 1. Tranexamic acid 2. No tranexamic acid | 180 | 1 year WOMAC pain: no significant difference Transfusion: control concern |
| Kusuma et al[68] USA, before 2013 1 hospital | Tourniquet, Esmarch bandage, electrocautery | 1. Thrombin infusion 2. No thrombin infusion | 80 | 6 months, 1 and 2 years KSS: no difference (p=0.45) |
| Napier et al[69] UK, 2003–2004 1 hospital | | 1. Passive flexion 2. Passive extension | 180 | 1 year OKS: no difference (p=0.27) Transfusion: extension concern |
| Sa-Ngasoongsong et al[64] Thailand, 2008–2009 1 hospital | Drain and compressive dressing | 1. Tranexamic acid 2. Control saline | 48 | 6 months WOMAC score: no difference (p=0.282) Transfusion: control concern |
| Sa-Ngasoongsong et al[65] Thailand, 2010–2011 1 hospital | Drain and compressive dressing | 1. Tranexamic acid 500 mg 2. Tranexamic acid 250 mg 3. Control saline | 135 | 1 year WOMAC score: no difference (p=0.42) Transfusions: control and 250 mg group concerns |
| **Denosumab** | | | | |
| Ledin et al[72] Sweden, 2012–2014 2 centres | | 1. Denosumab 2. Placebo | 50 | 1 and 2 years KOOS pain: no significant difference |
| **Continuous passive motion** | | | | |
| Bennett et al[74] Australia, 1997–2000 1 hospital | Physiotherapy | 1. Standard CPM 2. Early flexion CPM 3. No CPM | 147 | 1 year KSS: no significant difference |
| Ersözlü et al[73] Turkey, 2003–2004 1 hospital | Physiotherapy | 1. CPM low and increasing 2. CPM high and increasing 3. No CPM | 90 | 2 years KSS: no difference (p=0.67) |
| **Electrical stimulation** | | | | |

| Study, country, patient recruitment dates Setting | Treatment common to randomised groups | Intervention | Patients, n | Follow-up Group difference |
|---|---|---|---|---|
| Avramidis et al[75] Greece, 2005–2006 1 hospital | Physiotherapy | 1. Transcutaneous electric muscle stimulation 2. No treatment | 76 | 1 year SF-36 bodily pain: strong evidence favouring electrical stimulation (p<0.001). Mean difference (12.5/100) close to MCID (16.9/100). OKS/ KSS: no difference |
| Moretti et al[77] Italy, 2008–2010 1 hospital | Rehabilitation protocol | 1. Pulsed electromagnetic fields 2. No treatment | 30 | 6 months and 1 year VAS pain: some evidence favouring electrical stimulation (p<0.05). Mean difference (2.1/10) greater than MCID (16.1/100) Knee swelling: electrical stimulation concern |
| **Rehabilitation** | | | | |
| Li et al[79] China, 2015–2016 1 hospital | Standard rehabilitation | 1. Walking guidance and training 2. No treatment | 86 | 6 months VAS pain/ HSS score: some evidence favouring walking (both p<0.01). Mean VAS pain difference (2.4/100) greater than MCID (16.1/100) |
| Liebs et al[81] Germany, 2003–2004 4 hospitals | CPM, physiotherapy, postdischarge aquatic therapy | 1. Early aquatic therapy 2. Delayed aquatic therapy | 185 | 6 months, 1 and 2 years WOMAC pain: no difference (p=0.22 at 12 months) |
| Mahomed et al[82] Canada, 2000–2002 2 centres | Physiotherapy | 1. Multidisciplinary supported early discharge and home physiotherapy 2. Transfer to rehabilitation centre | 234 hip or knee replacement | 1 year WOMAC pain: weak evidence favouring supported discharge (p=0.08). Mean difference (4) less than MCID (8–9) |
| Wang et al[80] China, 2009–2010 1 centre | | 1. Wound closure in flexion 2. Wound closure in extension | 80 | 6 months VAS pain: no difference (p=0.64) |
| **Wound management** | | | | |
| Kong et al[71] South Korea, 2011 1 surgeon | Skin staples and closure strip | 1. Silicone gel 2. Petroleum gel | 100 | 6 months and 1 year VAS pain: no difference (6 months p=0.886, 1 year p=0.201) |
| **Anabolic steroids** | | | | |
| Hohmann et al[83] Australia, Before 2010 1 surgeon | CPM. Cold compression | 1. Intramuscular nandrolone injections 2. Saline injections | 10 | 6 and 9 months, 1 year KSS: some evidence favouring nandrolone (6 months p=0.04, 9 months p=0.06, 12 months p=0.03). Difference at 12 months (10.2) close to MCID (12.3) Bone mineral density: weak evidence favouring nandrolone |

ACB, adductor canal block; CPM, continuous passive motion; DN4, Douleur Neuropathique 4; FNB, femoral nerve block; HSS, Hospital for Special Surgery; KOOS, Knee injury and Osteoarthritis Outcome Score; KSS, Knee Society Score; LIA, local infiltration analgesia; MCID, minimal clinically important difference; NRS, numerical rating scale; OKS, Oxford Knee Score; PCA, patient-controlled analgesia; SF-36, Short Form 36 Health Survey; S-LANSS, Leeds Assessment of Neuropathic Symptoms and Signs Pain Scale; SNB, sciatic nerve block; VAS, visual analogue scale; VTE, venous thromboembolism; WOMAC, Western Ontario and McMaster Universities Osteoarthritis Index.

Similarly, there were no differences between single-shot SNB and PCA in WOMAC pain scale or VAS-rated pain at rest or during mobilisation, or between continuous SNB and PCA.

## Local anaesthetic infiltration

In six RCTs, treatment with LIA was investigated.

Three RCTs compared intraoperative LIA with placebo or no intervention. In one study, all 280 participants received FNB and PCA.[45] There was weak evidence that WOMAC pain scores were better in the LIA group at 6 months (p=0.063) but not at 12 months (p=0.107) when the difference in means of 3.8/100 was lower than the MCID of 8–9/100 reported by Ehrich and colleagues.[46] In another study, 56 patients received LIA including ketorolac, or saline placebo, and all received PCA.[47] At 1 year, mean differences and CIs provided weak evidence that OKS scores were better in the LIA group but the difference in means of 2.7/48 was less than the MCID of 4/48 reported by Beard and colleagues.[48] LIA before surgical incision was compared with placebo in one study with 120 participants.[49] None received FNB or PCA. There was weak evidence for a better KSS (function and knee score components) at 6 months in those receiving LIA (p=0.07) with a difference in means of 14.2/200 exceeding the MCID of 12.3/200 reported by Lee and colleagues.[50]

In one study, 51 participants received LIA intraoperatively, followed by PCA.[51] Those randomised to further postoperative catheter-delivered LIA with ketorolac, or saline placebo had similar VAS-rated pain at 6 and 12 months.

LIA delivered as an injection and postoperative infusion was compared with epidural PCA in one study with 222 patients.[52] There was no difference between groups in OKS at 12 months.

In one study of 100 participants, LIA with or without corticosteroid were compared.[53] All patients received PCA. At 2 years there was no difference in OKS between groups.

## Oral celecoxib

In one RCT, 44 participants received oral celecoxib or placebo,[54] as well as PCA. There were no differences between groups in KOOS or VAS-rated pain at 12 months.

## Ketamine or nefopam infusion

In one RCT, ketamine infusion, nefopam infusion and saline placebo were compared in 75 patients, all of whom received PCA.[55] VAS-rated pain on movement did not differ between groups at 12 months. For the Douleur Neuropathique 4 measure of neuropathic pain, there was some evidence favouring ketamine over placebo at 6 and 12 months (p=0.02), but overall, few patients reported neuropathic pain at 12 months.

## Pregabalin

Oral pregabalin was compared with placebo in one RCT with 240 participants.[56] All received LIA and PCA. At 6 months, there was some evidence for better NRS pain in patients receiving pregabalin compared with placebo (p=0.0176) but the difference in means of 0.54/10 was less than the MCID of 1/10 reported by Salaffi and colleagues.[57] No participants receiving pregabalin reported neuropathic pain when assessed using the S-LANSS, compared with 5.2% of those receiving placebo (p=0.014). Patients receiving pregabalin were more likely to be sedated and confused in the first 2 days after surgery.

## Tourniquet

Five studies with 399 participants explored tourniquet use to provide a bloodless field. Two studies each were from Australia and China, and one from Denmark. All were conducted at a single centre with participants recruited between 2008 and 2015. Sample sizes ranged from 20 to 150 participants, with a median of 65. The range of mean ages of participants in randomised groups was 66–71 years and in 3/5 studies, a majority of participants were women.

In three RCTs, participants received TKR with or without a tourniquet. In one study with 64 patients, a difference in KOOS pain favouring tourniquet use was not significant at 6 or 12 months.[58] In another study with 20 patients, the OKS was not significantly different between groups at 6 or 12 months.[59] There were three blood transfusions in the tourniquet group, compared with none in the 'no tourniquet' group. In the third study with 100 participants, VAS-rated pain and HSS scores were similar between groups at 6 months.[60] Six cases of wound ooze occurred in the tourniquet group.

In two RCTs, short and long-duration tourniquet use were compared. In one study with 65 participants, there was weak evidence based on graphical representation of means and CIs for improved OKS at 12 months in the long-duration group and the difference in means of 5/48[61] was greater than the MCID of 4/48. Adverse events were reported by 62% of participants receiving short-duration tourniquet compared with 38% in the long-duration group. The study was terminated early as 10 blood transfusions were required in the short-duration group compared with three in the long-duration group. In the second study with 150 participants, tourniquets were used in three different periods during surgery.[62] At 6 months, there were no differences between groups in HSS scores.

## Blood conservation

Seven studies with 829 participants evaluated strategies to limit blood loss after TKR. Two studies were from Thailand, and one each from China, France, South Korea, the UK and the USA. All were conducted at a single centre with participants recruited between 2003 and 2015 when stated. Sample sizes ranged from 48 to 180 participants, with a median of 106. One study had three trial arms. The range of mean ages of participants in randomised groups was 65–74 years and in all studies, a majority of participants were women.

## Tranexamic acid

Five RCTs evaluated tranexamic acid.

Tranexamic acid injections or infusions were compared with saline placebo or untreated control in four RCTs.[60 63–65] In all studies, control patients required more blood transfusions. In one study including 180 participants comparing intravenous tranexamic acid with untreated controls, there was no significant difference in WOMAC pain scores at 1 year.[63] In another study with 48 participants comparing intra-articular tranexamic acid injection with saline placebo, there was no significant difference in WOMAC scores at 6 months.[64] One study with 135 participants compared two intra-articular tranexamic acid doses and saline control.[65] There were no significant differences in WOMAC scores at 1 year. Intravenous and intra-articular tranexamic was compared with untreated controls in one study with 100 participants.[60] VAS-rated pain at 6 months was similar between groups, but there was strong evidence favouring tranexamic acid for HSS scores (p<0.001), although the difference in means of 1.4/100 was lower than the MCID of 8.3/100 reported by Singh and colleagues.[66]

In one study, continuous tranexamic acid infusion was compared with a single bolus in 106 patients.[67] There was no difference between groups in KSS at 6 months or blood loss.

## Thrombin infusion

In one RCT with 80 participants, thrombin infusion was compared with untreated control.[68] At 1 year there was no difference between groups in pain measured on the KSS.

## Flexion or extension

For blood management, operated knees were kept in passive flexion or passive extension after surgery in one RCT with 180 patients.[69] At 1 year, OKS was similar between groups. Transfusion requirement was greater in patients with passive extension.

## Compression bandage

One RCT conducted at a single UK centre with 49 participants recruited between 2013 and 2014 compared compression bandaging to reduce postoperative knee swelling with standard bandaging. The mean age of participants was about 69 years and a majority were women. OKS was similar in randomised groups at 6 months.[70]

## Wound management

One RCT with recruitment in 2011 at a single centre in South Korea evaluated a wound care strategy to limit postoperative scar pain. The mean age of participants was about 69 years and a majority were women. Investigators compared silicone gel application to the surgical scar with placebo in 100 participants.[71] There were no significant differences in VAS-rated pain at 6 and 12 months.

## Denosumab

One RCT evaluated use of the antiresorptive monoclonal antibody denosumab to promote bone healing. The study was conducted in two centres in Sweden with recruitment of 50 participants between 2012 and 2014. The mean age of participants was about 65 years and a majority were women. At 12 and 24 months there were no significant differences between groups in KOOS pain.[72]

## Continuous passive motion

Two RCTs with 237 participants evaluated use of CPM to minimise joint stiffness and improve range of movement. Studies were conducted in single centres in Australia and Turkey with participant recruitment between 1997 and 2004 and both had three trial arms. Sample sizes were 90 and 147 participants. The mean ages of participants in studies were about 63 and 72 years and a majority of participants were women. In one study, 90 participants were randomised to no CPM, CPM at low flexion from postoperative day 1 to day 7, or CPM at high flexion from postoperative day 3 to day 7.[73] There was no significant difference between groups in KSS at 2 years. In the other study, 147 participants were randomised to CPM with increasing range of movement from day 1 to day 6, early flexion CPM from day 0 to day 6, or no CPM.[74] There were no significant differences between groups in KSS at 12 months.

## Electrical stimulation

Two RCTs with 106 participants conducted in single centres in Greece and Italy evaluated electrical stimulation which is believed to have anti-inflammatory activity and limit muscle atrophy. Studies included 76 and 30 participants recruited between 2005 and 2010. The mean ages of participants were 71 and 70 years and in one study that reported it, a majority of participants were female.

In one study with 76 participants receiving transcutaneous electric muscle stimulation from postoperative day 2 for 6 weeks or no intervention, SF-36 bodily pain showed strong evidence for greater improvement at 1 year in the intervention group compared with control (p<0.001).[75] The difference in means of 12.5/100 was close to the MCID of 16.9/100 reported by Escobar and colleagues.[76] There were no differences in OKS or KSS scores. In another study with 30 participants, pulsed electromagnetic fields from postoperative day 7 were compared with untreated control.[77] At 12 months, there was some evidence that VAS-rated pain was lower in intervention patients compared with controls (p<0.05). The difference in means of 2.1/10 was greater than the MCID of 16.1/100 reported by Danoff and colleagues.[78] Knee swelling was common during the intervention.

## Rehabilitation

Four RCTs with 585 participants recruited between 2000 and 2016 evaluated features of early rehabilitation focusing on regaining range of movement, functional independence and improving mobility. Two studies were conducted at single centres in China and at two and four centres in Canada and Germany, respectively. Sample sizes ranged from 80 to 234 participants, with a median of 136.

The range of mean ages of participants in randomised groups was 68–78 years and in 3/4 studies, a majority of participants were women.

### Walking guidance and training

In one study, 86 participants were randomised to walking guidance and training from postoperative day 2 or no intervention further to standard rehabilitation.[79] At 6 months, there was some evidence that those receiving intervention had lower VAS-rated pain (p<0.01) and HSS score (p<0.01) than controls. The difference in mean VAS-rated pain of 2.4/10 was greater than the MCID of 16.1/100.

### Flexion or extension during knee closure

Targeting improved functional recovery, wound closure performed in 90° flexion was compared with wound closure in full extension in one study with 80 participants.[80] There was no difference between groups in VAS-rated pain at 6 months.

### Aquatic therapy

In one study with 185 participants, aquatic therapy commencing on postoperative day 6 was compared with aquatic therapy commencing on day 14.[81] Patients reported similar WOMAC pain at 12 and 24 months.

### Supported early discharge

In one study, early discharge supported by physiotherapist home visits and outpatient or self-directed physiotherapy was compared with 2 weeks of rehabilitation centre-based usual care.[82] The study included 234 individuals receiving TKR or total hip replacement. Compared with usual care, there was weak evidence that patients with early discharge had lower WOMAC pain scores at 12 months (p=0.08). The difference in means of 4 was less than the MCID of 8–9/100. Results were not presented separately but did not differ between patients with TKR or total hip replacement.

### Anabolic steroids

Searches identified one study of anabolic steroids to improve postoperative muscle strength conducted in one centre in Australia with recruitment of 10 participants before 2010. The mean age of participants was about 66 years and a minority were women. Participants received intramuscular nandrolone injections or saline from postoperative day 5 for 6 months. KSS results indicated some evidence for improvement in the intervention group compared with controls at 12 months (p=0.03).[83] The difference in means of 10.2/200 was close to the MCID of 12.3/200.

### DISCUSSION

Much research in TKR aims to identify treatments that facilitate a speedy recovery with minimal short-term pain. However, patients choose to have joint replacement for long-term pain relief and reduction in functional limitations. Thus, changes to perioperative care, supported by short-term RCT evidence, should be backed up with evidence about long-term effectiveness for reducing pain and reassurance that there are no long-term unfavourable consequences. To this end, we synthesised evidence from RCTs evaluating perioperative interventions which have considered their long-term effects on pain outcomes.

Consistent with its status as a key perioperative risk factor, a major focus of research into improving long-term pain after TKR has been through prevention of acute postoperative pain using multimodal analgesia. Our review provides good-quality evidence for a small benefit for intra-articular LIA injections, as previously shown in short-term studies,[31 84] oral pregabalin, oral opioids, and in relation to neuropathic pain, ketamine infusion. As well as potential benefits for reduced long-term pain, future studies will need to consider concerns associated with these interventions which may not have been identified in small studies including infection,[31] venous thromboembolism[36] and sedation.[56]

Nerve blocks are effective for managing perioperative pain[85] but we identified no long-term benefit. In single studies, there was no benefit for nefopam infusion, oral celecoxib or LIA with additional corticosteroid. Regarding future studies, standardisation of the multimodal regimen will allow evaluation of extra or alternative components in multiple studies in different settings. With such an approach, convincing evidence will accrue to guide multimodal pain management.

Some interventions targeted the prevention of adverse events and facilitation of early mobilisation. Tranexamic acid is highly effective in reducing blood transfusions during TKR[86] and we found no evidence that tranexamic acid affects long-term pain or, consistent with registry studies,[87 88] adverse events. Single RCTs of thrombin infusion and maintenance of knee in flexion to prevent blood loss showed no effect on long-term pain. Tourniquets improve intraoperative visualisation of the joint, reduce blood loss and facilitate cement fixation but are associated with nerve damage, delayed recovery, acute pain and need for analgesics.[89 90] The RCTs we identified showed no effects of tourniquet use on long-term pain.

As shown in a previous review,[91] there was no suggestion that CPM affects long-term pain. There was good-quality evidence for a small benefit for reduced long-term pain in patients receiving walking training, anabolic steroid injection, electrical stimulation and supported discharge.

For some interventions a direct mechanism is clear, but for others, reasons for long-term impact are less obvious. This may explain why, for example, no studies evaluated DVT prophylaxis with long-term follow-up excepting a small number reporting adverse events. However, treatments to prevent symptomatic DVTs which occur in about 1% of treated patients[92] also reduce the incidence of asymptomatic DVT observed in about 28% of treated patients[93] and this may have long-term benefits. Conversely, new anticoagulants are associated with

bleeding,[94] which may increase the risk of wound complications[95] and joint infection[96] which are associated with long-term pain.[97 98] More perioperative interventions with no information on long-term pain outcomes from RCTs are shown in figure 1.

Our study is limited by the lack of meta-analysis which was not appropriate due to intervention and outcome heterogeneity. In the context of perioperative pain management, this was noted previously.[84] Our approach to assessing the evidence was a narrative synthesis of studies with low risk of bias. While this may seem overly restrictive, Cochrane risk of bias assessment allows us to screen out studies with important issues that may affect the validity of results. The main potential source of bias was incomplete outcome assessment. Although studies with long-term follow-up are naturally at higher risk of missing data, we maintained a standard in this domain as it is recognised that research participants who do not complete follow-up assessments differ in outcomes from those with follow-up data and their inclusion could change the interpretation of results.[99]

Another limitation is that pain assessed with questionnaires does not take into account the effect of pain medications and assistive aids. About 58% of women and 40% of men report taking pain medications after TKR because of pain in the operated knee[100] and we must recognise that pain levels at follow-up without this treatment might be considerably higher. Even with treatment, around 20% of patients report chronic pain after TKR[10] and in the context of a blinded RCT we should expect to be able to identify effects of perioperative treatments.

We summarised p values to assess the strength of evidence but, as statistically strong evidence may not reflect clinically important results,[101] where possible we also compared effect sizes with MCIDs. Our review considered a diverse range of interventions at a specific time in the TKR pathway and, as we were unable to make clinical practice recommendations, we did not adopt the Grades of Recommendation, Assessment, Development and Evaluation system[102] for this review.

An alternative approach to the prevention of chronic pain after TKR is the individualisation of care based on pain phenotype, genetic, psychosocial and other factors.[103] An example of this might be the perioperative treatment only of individuals with neuropathic pain with pregabalin, as opposed to the non-stratified provision in the RCT of Buvanendran and colleagues.[56] In an RCT with pregabalin provided to patients with painful HIV neuropathy, while no overall benefit was seen, a group with hyperalgesia responded to pregabalin treatment.[104]

Our systematic review of perioperative interventions brings together evidence on interventions in the perioperative phase of the TKR pathway. There was good-quality evidence for some interventions of a small benefit for reduced long-term pain, and while not supportive of the inclusion of specific interventions in clinical practice, there are clearly areas that merit research. High-quality studies assessing long-term pain after perioperative interventions are feasible and necessary to ensure that patients with osteoarthritis achieve good long-term outcomes after TKR.

### Acknowledgements
We thank Dr Mario Moric for conducting additional analyses on the study of Buvanendran and colleagues.

**Contributors** ADB, JD, RGH and AWB contributed to the conception and design of the study. ADB, JD and VW undertook the systematic review. ADB and JD carried out the risk of bias assessments. ADB drafted the article with revisions by JD, VW, RGH and AWB. All authors approved the final version for publication.

**Funding** This article presents independent research funded by the National Institute for Health Research (NIHR) under its Programme Grants for Applied Research programme (RP-PG-0613–20001). This study was supported by the NIHR Biomedical Research Centre at University Hospitals Bristol NHS Foundation Trust and the University of Bristol.

**Disclaimer** The views expressed in this publication are those of the authors and not necessarily those of the NHS, the National Institute for Health Research or the Department of Health and Social Care.

**Competing interests** None declared.

**Patient consent for publication** Not required.

**Provenance and peer review** Not commissioned; externally peer reviewed.

**Data availability statement** All data relevant to the study are included in the article or uploaded as supplementary information.

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
