## [Reviewer comments · BMJ Open]

ARTICLE DETAILS

TITLE (PROVISIONAL)	Are peri-operative interventions effective in preventing chronic pain after primary total knee replacement? A systematic review
AUTHORS	Beswick, Andrew; Dennis, Jane; Gooberman-Hill, Rachael; Blom, AW; Wylde, Vikki

VERSION 1 – REVIEW

REVIEWER	Eske Kvanner Aasvang Copenhagen University, Rigshospitalet, Denmark
REVIEW RETURNED	22-Dec-2018

GENERAL COMMENTS	The authors are to be applauded for their extensive review of RCTs involving various perioperative interventions and their relation to persistent postoperative pain. Several interventions which may not normally be assumed related to pain are evaluated, including reduction of bleeding by tranexamic acid, which I also find to be a strength. The manuscript is well written, the tables easy to read, and the supplemental material extensive. The majority of interventions have very limited (many cases 1-5 studies) data and were too heterogeneous to allow pooling of data. This is methodologically sound, but I think the manuscript would benefit from a discussion reflecting more on the available literature and suggested mechanisms for pain, something the research group are renowned for. One of the main issues I think should be discussed when the potential benefits of the perioperative interventions are discussed, especially in the light of the relative small clinical effect (for instance ketamine), are adverse effect. For instance thrombosis in tranexamic acid (none according to the recent reviews), sedation (gabapentine, ketamine) I am also critical regarding the studies only assessing "neuropathic" pain, as from a patient's perspective, the distinction between "neuropathic" and "non-neuropathic" pain may not be important. An example is the Buvanandran Gabapentine study which although gaining much attention, has yet to produce data on the overall occurrence of pain, and not only neuropathic. I believe this and other studies with similar reporting, should be assessed more stringently in the bias assessment (supplemental material). Another issue is the potential effect of other interventions during the study period. Patients experiencing pain are expected to receive help including analgesics, some protocolized, but many not as they are over-the-counter. A similar topic, and very relevant due to the opioid epidemic, is the analgesic use at follow-up, which I don't see being addressed. It would be expected that relevant pain was treated during the 6-12 months leading up to the
---

	assessment, and often by opioids. Please adjust the discussion, or even better, the results accordingly Where there any studies looking at the effect of surgical technique? Patella eversion etc?
--	---

REVIEWER	Gwyn Lewis Auckland University of Technology, New Zealand
-----------------	--

REVIEW RETURNED	10-Jan-2019
-------------

GENERAL COMMENTS	Are peri-operative interventions effective in preventing chronic pain after primary total knee replacement? A systematic review Beswick et al. This systematic review investigated the impact of a range of perioperative interventions on long-term pain following TKA. The review included 44 studies but found limited high-quality evidence for any factors that may enhance long-term outcomes. The Introduction should include more recent and clearer indications of the risk factors for persistent pain following TKA, e.g., Harmelink et al. (2017), Lewis et al. (2015), Katz & Seltzer (2009). I am not sure I follow the categorisation of direct and indirect factors here. Given the general lack of predictive ability in determining what increases the risk for persistent pain, it is hard to know what might be a direct consequence or indirect. This also relates to the Discussion section on DVT prophylaxis, as I don't really see the point of this paragraph. It also should be recognised that most of the risk factors for persistent pain following TKA are present pre-operatively, e.g. psychological factors, and are not specifically related to surgical and analgesic procedures. I wondered the value of including studies that do not directly address persistent pain, but rather are designed to optimise other outcome, such as blood loss. Is it really informative to include any peri-operative interventions? Perhaps it could be stated more overtly how these interventions/outcomes might be related to persistent pain. It seems a little odd to evaluate an intervention that is not designed to address the outcome measure of interest. The Results section is missing some key information on the studies. An integrated summary of participant characteristics, outcome measures used, time periods of assessment, study quality, as well as the interventions and the study findings is required. At present, the Results section only describes the interventions and outcomes, and this could be done in a more integrated and succinct manner. The opening paragraphs of the Discussion are not really relevant – it seems to take a long time to get to the point. I also found it odd that the authors seem to make recommendations for further research rather than conclusions related to the specific interventions reviewed. It would be more meaningful to know, for example, that there is low/moderate/high quality evidence that LIA injections are/are not effective in reducing persistent pain. They authors state that they are not supportive of interventions to optimise outcomes, but is there some evidence that certain interventions do not change outcomes? This is just as important. I also think there should be some discussion of how the interventions included align with the known risk factors for persistent pain. What types of interventions appear to be lacking?
--

	Minor/specific comments Abstract, lines 10-11. It is not clear what specific factors of peri-operative care are associated with persistent pain. There are known risk factors – which ones are related to peri-operative care? Interpretation. I don't think this provides an informative interpretation at all. It provides a vague summary of what was done. Page 4, lines 37-38. This is an odd sentence to be placed in here. I do not see how it relates to the sentences around it regarding treatments in the peri-operative period. Page 5, 1st paragraph. I do not see the relevance of the paragraph on patient and public involvement. Page 5, Eligibility Criteria. This section would be better in paragraph format. The inclusion/exclusion criteria need to be more specific, e.g., “predominantly for osteoarthritis” and “early stages of recovery” and vague and not at all clear. Were there date range/English language requirements? Search strategy. Why was “review” included in the search terms when reviews were excluded? Figure 1. Why is there an initial search and a follow-up search? This is not mentioned in the Methods. Page 20, lines 16-18. This sentence does not make sense. Page 20, final paragraph. I am not sure why the section on studies with no long-term outcome is included. If these studies did not meet the inclusion criteria, they should not be included or described.
--	--

VERSION 1 – AUTHOR RESPONSE

Reviewer: 1

The authors are to be applauded for their extensive review of RCTs involving various perioperative interventions and their relation to persistent postoperative pain. Several interventions which may not normally be assumed related to pain are evaluated, including reduction of bleeding by tranexamic acid, which I also find to be a strength.

The manuscript is well written, the tables easy to read, and the supplemental material extensive.

Thank you for these supportive comments, our aim was to review all aspects of care in the peri-operative setting in the belief that any treatment can have an impact on long-term pain.

The majority of interventions have very limited (many cases 1-5 studies) data and were to heterogeneous to allow pooling of data. This is methodologically sound, but I think the manuscript would benefit from a discussion reflecting more on the available literature and suggested mechanisms for pain, something the research group are renowned for.

We agree that a discussion of pain mechanisms is appropriate and have added to the Background section to address this.

“The mechanisms that influence the development of chronic pain after total knee replacement may be biological, mechanical and psychosocial. Biological causes include the sensitising impact of long-term pain from osteoarthritis[11,12], inflammation, infection and localised nerve injury[13]. Mechanical causes include altered gait, prosthesis loosening, and effects on ligaments[14,15]. Psychological factors including depression and catastrophizing may also influence outcomes[16-19].”

One of the main issues I think should be discussed when the potential benefits of the perioperative interventions are discussed, especially in the light of the relative small clinical effect (for instance ketamine), are adverse effect. For instance thrombosis in tranexamic acid (none according the recent reviews), sedation (gabapentine, ketamine)

Our focus has been on pain outcomes and adverse events reported in the RCTs we identified. As the reviewer notes, this pragmatic approach does not consider all evidence relating to adverse events. We have added information from more general research into adverse events to paragraph 3 of the discussion.

As well as potential benefits for reduced long-term pain, future studies will need to consider concerns associated with these interventions which may not have been identified in small studies including infection[32], venous thromboembolism[40] and sedation[54].

I am also critical regarding the studies only assessing “neuropathic” pain, as from a patients perspective, the distinction between “neuropathic” and “non-neuropathic” pain may not be important. An example is the Buvanendran Gabapentine study which although gaining much attention, has yet to produce data on the overall occurrence of pain, and not only neuropathic. I believe this and other studies with similar reporting, should be assessed more stringent in the bias assessment (supplemental material).

We agree that from a patient perspective, chronic pain is the problem, irrespective of whether we classify it as nociceptive or neuropathic. In hindsight, we agree that it was inappropriate to consider exclusively neuropathic pain as an appropriate outcome. Thus, we requested information from Buvanendran and colleagues on whether overall pain outcome measured at 6 months was available. This week we received this data and have included it in the tables and article and added an acknowledgement.

Another issue is the potential effect of other interventions during the study period. Patients experiencing pain are expected to receive help including analgesics, some protocolized, but many not as they are over-the-counter. A similar topic, and very relevant due to the opioid epidemic, is the analgesic use at follow-up, which I don't see being addressed. It would be expected that relevant pain was treated during the 6-12 months leading up to the assessment, and often by opioids. Please adjust the discussion, or even better, the results accordingly Where there any studies looking at the effect of surgical technique? Patella eversion etc?

Our review outcome of long-term pain is clearly influenced by the treatment patients receive for pain. We have added a section to the discussion covering this.

“Another limitation is that pain assessed with questionnaires does not take into account the effect of pain medications and assistive aids. About 58% of women and 40% of men report taking pain medications after TKR because of pain in the operated knee[105] and we must recognise that pain levels at follow up without this treatment might be considerably higher. Even with treatment, around 20% of patients report chronic pain after TKR[10] and in the context of a blinded RCT we should expect to be able to identify effects of peri-operative treatments.”

In this review we focused on non-surgical interventions during the knee replacement pathway. In a new review, we will be looking at knee implant design and surgical factors and long-term clinical and patient outcomes.

Reviewer: 2

This systematic review investigated the impact of a range of perioperative interventions on longterm pain following TKA. The review included 44 studies but found limited high-quality evidence for any factors that may enhance long-term outcomes.

The Introduction should include more recent and clearer indications of the risk factors for persistent pain following TKA, e.g., Harmelink et al. (2017), Lewis et al. (2015), Katz & Seltzer (2009).

As suggested, we have added a brief section to the Background citing research into risk factors for chronic pain. We now cite Lewis 2015 and Blom 2016 as reviews of pre-operative risk factors. The focus of reviews has been mainly on pre-operative factors but as yet we are not aware of successful targeting of pre-operative risk factors to prevent chronic pain and note this.

“Much research has focused on pre-operative predictors of outcomes and these include pain intensity, presence of widespread pain, anxiety, depression and catastrophizing.[10,18] However attempts to target or modify pre-operative care have shown little long-term benefit regarding chronic pain or other long-term patient outcomes[10,19-21].”

Stratified care based on pre-operative risk factors may be the way forward, and we have added a brief section to the discussion addressing this. However, we consider the possibility that features of care in the peri-operative period may have consequences for pain outcomes in the long-term and need to be researched just as pre-operative interventions do. We have added text on this which also addresses an issue on pain outcome discussed by Reviewer 1.

“An alternative approach to the prevention of chronic pain after TKR is the individualisation of care based on pain phenotype, genetic, psychosocial and other factors[108]. An example of this might be the peri-operative treatment only of individuals with neuropathic pain with pregabalin, as opposed to the non-stratified provision in the RCT of Buvanendran and colleagues[53]. In an RCT with pregabalin provided to patients with painful HIV-neuropathy, while no overall benefit was seen, a group with hyperalgesia responded to pregabalin treatment[109].”

I am not sure I follow the categorisation of direct and indirect factors here. Given the general lack of predictive ability in determining what increases the risk for persistent pain, it is hard to know what might be a direct consequence or indirect.

We agree that it is hard to know what might be a direct or indirect consequence of an intervention. To explain this better, and to emphasise that chronic pain is an adverse event, we have altered the background:

“Any treatment in the peri-operative period including pain management, blood conservation, deep vein thrombosis (DVT) and infection prevention, and inpatient rehabilitation could potentially affect patient recovery and chronic pain, either directly or indirectly. Direct mechanisms may be through prevention of nerve damage[25], post-thrombotic syndrome[26], reperfusion injury[27] and articular bleeding[28]. For other treatments, pathways leading to long-term pain may be indirect, possibly being mediated through increased risks of adverse events[29]. Irrespective of mechanism, chronic pain is a highly prevalent adverse event after TKR and should be considered along with infection, DVT and other complications in the safety profile of interventions.”

This also relates to the Discussion section on DVT prophylaxis, as I don't really see the point of this paragraph.

We included this paragraph as an example of an intervention where the mechanism relating to long-term pain may be “indirect” and thought it might interest readers of the article. We have now made it clear that it is an example. If the reviewers and editors consider the paragraph does not add anything to the discussion, we will remove it.

It also should be recognised that most of the risk factors for persistent pain following TKA are present pre-operatively, e.g. psychological factors, and are not specifically related to surgical and analgesic procedures.

In the Background section we have now described the key pre-operative risk factors for poor outcomes. Evidence is more convincing than for postoperative patient-related risk factors.

Unfortunately, modification of pre-operative risk factors or targeting of care has, as yet, not yielded an intervention with convincing evidence of benefit on chronic pain.

“The mechanisms that influence the development of chronic pain after total knee replacement may be biological, mechanical and psychosocial. Biological causes include the sensitising impact of long-term pain from osteoarthritis[11,12], inflammation, infection and localised nerve injury[13]. Mechanical causes include altered gait, prosthesis loosening, and effects on ligaments[14,15]. Psychological factors including depression and catastrophizing may also influence outcomes[16-19]. Much research has focused on pre-operative predictors of outcomes and these include pain intensity, presence of

widespread pain, anxiety, depression and catastrophizing.[10,20] However, attempts to target or modify pre-operative care have, as yet, shown no benefit regarding chronic pain or other long-term patient outcomes[10,21-23].”

I wondered the value of including studies that do not directly address persistent pain, but rather are designed to optimise other outcome, such as blood loss. Is it really informative to include any perioperative interventions? Perhaps it could be stated more overtly how these interventions/outcomes might be related to persistent pain. It seems a little odd to evaluate an intervention that is not designed to address the outcome measure of interest.

This review is part of a broader 5-year programme of work, with an overarching theme of the prevention of long-term pain after knee replacement. We have looked previously at predictive factors for long-term pain, both pre-operative and post-operative factors. The next step in our research relates to modification of risk factors with diverse interventions as reported here in the peri-operative context.

Our decision to include the whole range of non-surgical interventions is because chronic pain is a major adverse event after knee replacement and of key importance to patients. Some interventions have a clear mechanism. We agree that a relationship between some interventions and chronic pain is somewhat tenuous, for example use of tranexamic acid. However, patients need reassurance that their care in hospital is not associated with adverse events, the most widely experienced being chronic pain.

We hope our modified Paragraph 5 from the Background addresses this.

The Results section is missing some key information on the studies. An integrated summary of participant characteristics, outcome measures used, time periods of assessment, study quality, as well as the interventions and the study findings is required. At present, the Results section only describes the interventions and outcomes, and this could be done in a more integrated and succinct manner.

We agree that information on study and patient characteristics is needed. We have added details of patient characteristics into the Results section Paragraph 2 and in the first paragraph of each intervention results: indication; intervention focus; primary pain outcome measure; follow up; adverse event reporting; country; number of centres; recruitment dates; sample sizes; number of trial arms; mean age; and percentage women.

Regarding study quality, we only included studies considered to be at low risk of bias and have now emphasised this in Paragraph 1 of the Results

The opening paragraphs of the Discussion are not really relevant – it seems to take a long time to get to the point.

We included the brief description of the value of systematic reviews and reason for the study as BMJ Open is a general medical journal. If the Editors feel these paragraphs are not relevant we can remove them.

I also found it odd that the authors seem to make recommendations for further research rather than conclusions related to the specific interventions reviewed. It would be more meaningful to know, for example, that there is low/moderate/high quality evidence that LIA injections are/are not effective in reducing persistent pain. They authors state that they are not supportive of interventions to optimise outcomes, but is there some evidence that certain interventions do not change outcomes? This is just as important. I also think there should be some discussion of how the interventions included align with the known risk factors for persistent pain.

We have not used the GRADE approach in this review and have acknowledged this.

Our interpretation of the evidence is that further research is required on long-term benefits of intra-articular LIA injections, oral opioids, and in relation to neuropathic pain, ketamine infusion, oral pregabalin, walking training, anabolic steroid injection, electrical stimulation and supported discharge. This conclusion (further research is required) is based on the encouraging but by no means definitive results of a small number of RCTs:

Intra-articular LIA injections – evidence was consistent in 3 studies (with no post-operative LIA delivery) but effect sizes were small

Oral opioids – benefit for one pain measure during activity but concern over venous thromboembolism
Ketamine infusion – benefit for neuropathic pain but not overall pain
Oral pregabalin – some evidence of small benefit for overall and neuropathic pain
Walking training – some evidence for benefit greater than MCID, but only in one study
Anabolic steroid injection – some evidence for benefit similar to MCID, but only in one small study
Electrical stimulation – two different interventions showed benefit similar or greater to MCIDs in single studies, and one was small
Supported discharge – weak evidence for benefit less than the MCID

What types of interventions appear to be lacking?

We summarised the interventions which have been evaluated in RCTs but with no robust evaluation at the end of the Results section. We have also added a brief section to the discussion highlighting the stratified care approach where risk factors are used to target care – this may for example be relevant for interventions like pregabalin.

Minor/specific comments

Abstract, lines 10-11. It is not clear what specific factors of peri-operative care are associated with persistent pain. There are known risk factors – which ones are related to peri-operative care?

Specific examples we note in the Abstract are nerve damage or complications. We have attempted to clarify this:

“Objectives

For many people with advanced osteoarthritis, total knee replacement (TKR) is an effective treatment for relief of pain and improvement of function. Features of peri-operative care may be associated with the adverse event of chronic pain six months or longer after surgery; effects may be direct, e.g. through nerve damage or surgical complications, or indirect through increasing risks of adverse events. The objective of this systematic review is to evaluate whether non-surgical peri-operative interventions prevent long-term pain after TKR.”

Interpretation. I don't think this provides an informative interpretation at all. It provides a vague summary of what was done.

We have now focused this on the findings of the review.

“To prevent chronic pain after TKR, peri-operative interventions including components of multimodal analgesia, early rehabilitation and supported discharge, electrical stimulation and anabolic steroids show promise that merits further research. Tranexamic acid use is not associated with chronic pain but the long-term consequences of many widely researched treatments have not been reported.”

Page 4, lines 37-38. This is an odd sentence to be placed in here. I do not see how it relates to the sentences around it regarding treatments in the peri-operative period.

We agree and psychological factors are now included in the new risk factors section

“The mechanisms that influence the development of chronic pain after total knee replacement may be biological, mechanical and psychosocial. Biological causes include the sensitising impact of long-term pain from osteoarthritis[11,12], inflammation, infection and localised nerve injury[13]. Mechanical causes include altered gait, prosthesis loosening, and effects on ligaments[14,15]. Psychological factors including depression and catastrophising may also influence outcomes[16-19]. Much research has focused on pre-operative predictors of outcomes and these include pain intensity, presence of widespread pain, anxiety, depression and catastrophizing.[10,20] However attempts to target or modify pre-operative care have, as yet, shown little long-term benefit regarding chronic pain or other long-term patient outcomes[10,21-23].”

Page 5, 1st paragraph. I do not see the relevance of the paragraph on patient and public involvement. BMJ Open now request that authors provide a Patient and Public Involvement statement in the methods section. PPI was important in developing the project and our PPI group will advise on dissemination of results.

Page 5, Eligibility Criteria. This section would be better in paragraph format. The inclusion/exclusion criteria need to be more specific, e.g., “predominantly for osteoarthritis” and “early stages of recovery” and vague and not at all clear. Were there date range/English language requirements?

We have changed the format of the Eligibility criteria as suggested. We have added in that searches were from database inception. There were no language restrictions.
 We have changed “predominantly” to “at least 75% of patients”.
 Also, the “early stages of recovery” are more clearly defined “Pharmacological or non-pharmacological interventions commenced in the peri-operative setting with “peri-operative” reflecting the time from hospital admission to immediately post-discharge.”
 Search strategy. Why was “review” included in the search terms when reviews were excluded?
 We searched for reviews to allow us to hand search reference lists as a possible source of extra relevant RCTs
 Figure 1. Why is there an initial search and a follow-up search? This is not mentioned in the Methods. The review was time consuming and we thought it important to update the searches to make the review more reflective of contemporary evidence. As the PROSPERO protocol stated the initial search date, we felt it appropriate to report both dates. If BMJ Open editors are in agreement we are happy to just report the final search date to avoid confusion.
 Page 20, lines 16-18. This sentence does not make sense.
 We apologise about this and we have now rewritten the sentence.
 “In one study with 185 participants, aquatic therapy commencing on post-operative day six was compared with aquatic therapy commencing on day 14[72].”
 Page 20, final paragraph. I am not sure why the section on studies with no long-term outcome is included. If these studies did not meet the inclusion criteria, they should not be included or described. It is true that this was not specified in PROSPERO. It is a commentary on the PRISMA flow diagram which summarises our findings from all RCTs during the screening process. To clarify that our searches identified all RCTs of interventions in the peri-operative setting, we have added briefly to the first paragraph of results.
 “Figure 1 shows review progress and reasons for exclusion. Of 1515 RCTs of interventions in the peri-operative setting, 1385 had no long-term follow up. Peri-operative interventions with follow up of ≥six months were evaluated in 130 RCTs of which 76 reported a pain outcome or score with a pain component.”
 Editor Comments to Author:
 - Please give the names of the databases searched in the abstract.
 Database names added
 - Please change the 'Interpretations' section to a 'Conclusions' section in the abstract.
 Interpretations changed to Conclusions
 - Please include the start search date in the methods section - was this from inception?
 “from database inception” added
 - Please include Figure legend at the end of your main manuscript.
 Figure 1 Legend added

VERSION 2 – REVIEW

REVIEWER	Eske Kvanner Aasvang Copenhagen University, Denmark
REVIEW RETURNED	29-Apr-2019

GENERAL COMMENTS	My comments have been adressed satisfactory, and the added information has made the text more baælanced and I am impressed that the authors managed to get the data on the no-neuropathic pain from the Buvenandran study!
--

REVIEWER	Gwyn Lewis Auckland University of Technology, New Zealand
REVIEW RETURNED	17-Apr-2019

GENERAL COMMENTS	The authors have much improved the manuscript and addressed most of my concerns. I have a few remaining. My biggest concern is in the discussion of the findings and the recommendations made. While a GRADE approach was not used, all studies included in the review were high quality RCTs, while the evidence within the Results section is additionally classified by P values. These factors alone are useful indicators of the quality of evidence presented. Stating “encouragement for further research” provides no useful information on the interventions. Stating, for example, that there is high quality evidence for a small benefit (not clinically meaningful) is far more useful. We are left not knowing if there was conflicting evidence, minimal number of studies, good evidence but there were adverse events, or consistent weak evidence, for example. This is the main point of the review and the conclusions need to be much more informative and clearer. I still find the opening paragraphs of the Discussion largely irrelevant. In relation to the comment on the types of interventions that are lacking, I was not referring to a stratified care approach. Rather, a series of risk factors are outlined in the Introduction, then a series of interventions presented in the Results. I am wondering if the interventions align to the known risk factors. It would definitely avoid confusion if the search strategy was just presented as the final search date. I still unclear on why the paragraph in the Results on studies with no long-term follow-up is included.
--

VERSION 2 – AUTHOR RESPONSE

We thank the reviewers for their further consideration of our article.

As noted by Reviewer 2, in places our original Abstract and Discussion section text provided no useful information and we have altered this as suggested. We have also added in the number of studies for each intervention in the abstract results. Small changes have been made to the abstract text to keep it under 300 words.

The first paragraph of the Discussion is removed as requested by Reviewer 2.

Regarding the alignment of peri-operative risk factors with interventions, we have modified the discussion so that each group of interventions has a lead-in relating to peri-operative risk factors.

We have kept the Results paragraph on studies with no long-term follow up as it may encourage researchers to undertake longer-term follow up of RCTs in the peri-operative setting for adverse events including long-term pain.

The search strategy is now consistently reported as up to February 2018.

We agree with Reviewer 1 that after incorporation of the reviewers' comments, our article is improved and more balanced. Following up the suggestion of Reviewer 1, we were really pleased to be able to include the overall pain data from the study of Buvenandran, co-author Dr Moric was very helpful in providing new analysis.

VERSION 3 – REVIEW

REVIEWER	Gwyn Lewis Auckland University of Technology
REVIEW RETURNED	09-Aug-2019

GENERAL COMMENTS	Review is a little tidier but I am confused on a couple of the conclusions. It looks like 3 out of the 4 studies that compared LIA to placebo did not show a statistically significant difference between groups, and in the remaining study the difference was less than clinically useful (P-value not presented). Two studies comparing LIA with other interventions found no differences. From this, I am not clear why there is good-quality evidence for LIA. Similarly, in the one study involving ketamine there was no significant difference in pain between groups so I am unclear why there is good quality evidence for this either. I am not sure why tranexaminic acid is specifically mentioned in the Abstract when there were many interventions that showed no effect. Surely it is an important finding that for most of the interventions, there was good quality evidence of no effect. The studies with no long-term outcomes are not part of the review. The absence can be commented upon in the Discussion, but they should not be presented in the Results (or Abstract). The Conclusion of the Abstract seems to be a re-wording of the Results section. Page 4, lines 20-24. I suggest removing the word "cause" from these statements and using something like "influence" instead. Page 7, line 31. 75% or more had osteoarthritis?
--

VERSION 3 – AUTHOR RESPONSE

We thank Reviewer 2 for these suggestions which we have addressed in the revised article.

We have made the LIA results section clearer. There were 3 studies comparing LIA with placebo/ no intervention. The other three studies compared LIA provided after wound closure through a catheter with placebo, LIA versus an alternative pain management modality, and LIA with or without corticosteroid. Thus we emphasise that the small benefit was seen for intra-operative LIA versus placebo or no intervention.

Regarding statistical significance, as recommended by Sterne and Davey Smith, we avoided describing results as statistically significant and considered p-values ≤ 0.001 as "strong" evidence of effectiveness. We considered p-values 0.001-0.05 "some" evidence, and p-values 0.05-0.1 "weak" evidence.

The results and conclusions sections of the abstract are rewritten to focus on the good quality interventions with long-term pain outcomes.

Regarding interventions that we identified with no data, we have removed them from the abstract and results section and make only a brief note in the Results section that these are listed in Figure 1. We have left DVT prophylaxis as an example in the discussion.

We have altered the sentences on Page 4 and 7 to make them read more clearly.

As noted by Reviewer 2, the Abstract conclusion was inadequate and we have changed this to:

“To prevent chronic pain after TKR, several peri-operative interventions show benefits and merit further research. Good quality studies assessing long-term pain after peri-operative interventions are feasible and necessary to ensure that patients with osteoarthritis achieve good long-term outcomes after TKR.”

VERSION 4 – REVIEW

REVIEWER	Gwyn Lewis Auckland University of Technology
REVIEW RETURNED	16-Aug-2019
GENERAL COMMENTS	The authors have addressed all my comments.